# Association between Intestinal Microecological Changes and Atherothrombosis

**DOI:** 10.3390/microorganisms11051223

**Published:** 2023-05-06

**Authors:** Xinyu Zhuo, Hui Luo, Rumei Lei, Xiaokun Lou, Jing Bian, Junfeng Guo, Hao Luo, Xingwei Zhang, Qibin Jiao, Wenyan Gong

**Affiliations:** 1Department of Clinical Medicine, Affiliated Hospital of Hangzhou Normal University, Hangzhou Normal University, Wenzhou Road, Gongshu District, Hangzhou 310000, China; 2019211301026@stu.hznu.edu.cn (X.Z.); hluo@hznu.edu.cn (H.L.); 2019211301027@stu.hznu.edu.cn (R.L.); 2019211301092@stu.hznu.edu.cn (X.L.); 2018211102041@stu.hznu.edu.cn (J.B.); 2019211301072@stu.hznu.edu.cn (J.G.); 2020211301005@stu.hznu.edu.cn (H.L.); xwzhang@hznu.edu.cn (X.Z.); 2Hangzhou Institute of Cardiovascular Disease, Hangzhou 310000, China

**Keywords:** intestinal microecology, atherosclerosis, TMAO, SCFAs, atherosclerotic plaque

## Abstract

Atherosclerosis (AS) is a chronic inflammatory disease of large- and medium-sized arteries that causes ischemic heart disease, strokes, and peripheral vascular disease, collectively called cardiovascular disease (CVD), and is the leading cause of CVD resulting in a high rate of mortality in the population. AS is pathological by plaque development, which is caused by lipid infiltration in the vessel wall, endothelial dysfunction, and chronic low-grade inflammation. Recently, more and more scholars have paid attention to the importance of intestinal microecological disorders in the occurrence and development of AS. Intestinal G-bacterial cell wall lipopolysaccharide (LPS) and bacterial metabolites, such as oxidized trimethylamine (TMAO) and short-chain fatty acids (SCFAs), are involved in the development of AS by affecting the inflammatory response, lipid metabolism, and blood pressure regulation of the body. Additionally, intestinal microecology promotes the progression of AS by interfering with the normal bile acid metabolism of the body. In this review, we summarize the research on the correlation between maintaining a dynamic balance of intestinal microecology and AS, which may be potentially helpful for the treatment of AS.

## 1. Introduction

### 1.1. Intestinal Microecology

Intestinal microecology is a dynamic and balanced system consisting of four parts: the intestinal anatomy, mucosal secretions, food intake, and microorganisms colonizing the intestine [1]. The human intestine, a part of the digestive tract, is divided into the small intestine and large intestine according to its morphology, function, and structure. The wall of the small intestine consists of a mucosal layer, sub-mucosal layer, muscular layer, and outer membrane, and the mucosal epithelium is composed of absorptive cells, cup cells, and Paneth cells, which make the small intestine the main part of nutrient absorption in the human body. The wall of the large intestine is similar to the wall of the small intestine, but its epithelial lamina propria contains dense colonic glands, which are the main sites for absorbing water from food residues and forming feces [2,3,4,5] (Figure 1). The intestinal microbiota consists of over 100 trillion microorganisms, with bacteria accounting for 90% of the total intestinal microorganisms, primarily from the Bacteroides and Firmicutes phyla [6]. Other intestinal microflora, such as Fusobacteria, Bacteroidetes, and Verrucomicrobiaphylum, are present in much smaller quantities [7].

The structure and function of intestinal flora vary across populations and are primarily influenced by the hosts’ diet, age, and living environment. The proportion of intestinal microflora varies among different age groups, with the bifidobacteria-dominated microbiota in infants shifting to a microbiota dominated by *Bacteroidetes* and *Firmicutes* in adults, while the relative proportion of the *phylum Bacteroides* dominates in the elderly [8]. Diet structure can also change the colonization environment of normal intestinal flora, affecting their growth, development, and reproduction, such as a high-fat diet increases the proportion of thick-walled bacteria phylum and decreases the proportion of bacteriophage phylum [9]. Antibiotic use leads to a downregulation of the density of the normal flora which is sensitive to antibiotics, while the non-sensitive flora proliferates and becomes dominant, thus showing a dysbiosis of microecology. With the development of molecular tools and technologies, such as macroeconomics, metabolomics, lipidomic, and Meta transcriptomics, it has been found that alterations in the intestinal flora are associated with obesity, type 2 diabetes, hepatic steatosis, intestinal diseases (IBDs), and various cancers. This discovery may provide a new way of thinking for the treatment of associated diseases [10] (Table 1). Furthermore, it has been demonstrated that gut microecology is indispensable in human immune regulation, metabolism, and food catabolism and metabolism [10].

### 1.2. Atherosclerosis

Atherosclerosis (AS) is a pathological process characterized by chronic inflammatory changes, lipid accumulation, fibrous tissue proliferation, and calcium deposition in the arterial vessel wall. This condition is accompanied by intraplaque hemorrhage, plaque rupture, and local thrombosis, which are the main causes of ischemic heart disease (IHD). According to the Global Burden of Disease study, the world prevalence of coronary heart disease (CHD) has increased from 100 million cases in 1990 to 180 million in 2019, with AS being the leading cause of CHD [11]. The concept of AS was first introduced in 1904 by the German pathologist Marchand, and subsequently, Russian pathologist Nikolai Anitschkow successfully induced rabbit AS models by feeding them cholesterol-rich sunflower oil [11]. AS is a multifactorial process that develops due to imbalances in lipid metabolism, inflammation, atherothrombosis, and changes in blood pressure. Inflammatory response, which is a major influencing factor, interacts with unbalanced lipid metabolism and other traditional risk factors that jointly contribute to the development of AS [12]. Low-density lipoprotein (LDL) is a particle encapsulated by Apo B that transports cholesterol to peripheral tissue cells, and AS occurs when the concentration of LDL is high [13,14,15]. These lipid particles can readily cross the intima and bind to chondroitin sulfate proteoglycans in the arterial wall matrix, leading to the deposition of cholesterol and cholesterol esters in the intima, which causes hyperplasia of arterial tissue and thickening and hardening of the arterial well [16].

The loss of elastic fibers in arteries and the thickening of small arterial walls are thought to be closely related to AS [17]. As atherosclerosis develops, the compliance of the arterial wall gradually decreases, and the shear forces generated by blood flow are highly susceptible to damage in the vascular endothelial cells, which leads to the adhesion of monocytes in the blood to the endothelium, and subsequent migration of monocytes to the subendothelial, where they differentiate into macrophages and then transform into foam cells after phagocytosis of lipid particles [17,18,19]. Clinical complications of atherosclerotic plaques include thrombosis caused by fibrous cap fracture or surface erosion of the plaque, which prevents perfusion, causes an acute ischemic syndrome, and promotes plaque healing and progressive intimal thickening. The combination of both will readily lead to further aggravating arterial stenosis, which leads to the limitation of perfusion, eventually resulting in ischemic tissue necrosis [20]. The intestinal flora analysis in patients with hyperlipidemia and hypercholesterolemia combined with colon cancer showed alterations in intestinal microbial composition, as evidenced by increased *Escherichia*/*Shigella* ratios and abundance of *Streptococcus*, as well as a decreased mass of *Clostridium XIVa* and *Clostridium tumefaciens* [21]. Jie et al. have conducted a metagenome-wide association study on the stools of 218 atherosclerotic patients and 187 healthy controls and found that increased *E. coli* and *Streptococcus* in the gut microbiome of atherosclerotic patients affected the transport or metabolism of molecules critical to cardiovascular health [22]. In recent years, the association between intestinal microecological alterations and AS development has gained increased attention [18,23,24,25,26]. Microbial metabolites represented by trimethylamine n-oxide (TMAO) have become important mediators in the development of atherosclerosis, which serve as substrates required for biochemical reactions in the organism and are involved in regulating intestinal homeostasis and influencing the development of atherosclerosis, thus yielding a wide range of potential therapeutic targets. In this review, we detail how changes in gut microecology affect the development of AS in terms of gut microbial composition, metabolites of gut microbes such as TMAO and SCFAs, cellular components of gut flora, and interference of gut microbes with bile acid metabolism in the body, and provide new ideas for the treatment and prevention of AS.

### 1.3. Intestinal Microecology and AS

#### 1.3.1. Intestinal G^−^ Bacterial Cell Wall Lipopolysaccharide (LPS) and AS

Endotoxin, also known as LPS, is present in the outer wall of the cell wall of Gram-negative bacteria and consists of O antigen, core polysaccharide, and lipid A. Typically, it is released during bacterial cell wall disruption or bacterial division, triggering a robust immune response in the host that results in acute infection symptoms, such as fever, sepsis, infectious shock, and respiratory distress syndrome [27,28]. Additionally, LPS is closely associated with other chronic diseases in the host that involve immune responses, and lipid-A plays a major immunostimulatory role among them [29].

Several studies have demonstrated that individuals at high risk of AS exhibits increased levels of LPS and its binding protein [25,30,31]. LPS, which is a component of the outer membrane of Gram-negative bacteria, triggers Toll-like receptor 4 (TLR4) activation, thereby promoting the development of AS through the activation of the body’s inflammatory response [25]. LPS can bind with lipopolysaccharide-binding protein (LBP) and either high-density lipoprotein-2 (HDL2) or chylomicrons, allowing it to reach the liver via the portal vein or the body’s circulation via the lymphatic system [32]. Targeting intestinal-derived oxidized phospholipids may reduce the uptake of LPS at the intestinal level, thus limiting the inflammatory response that promotes AS [33]. LPS stimulates inflammatory responses in a TLR4-dependent manner and plays an important role in pro-leukocyte activation and release of inflammatory mediators, dendritic cell maturation and migration, macrophage autophagy, increase in reactive oxygen species (ROS) and reactive nitrogen species (RNS) formation, and exertion of a wide range of effects on vascular endothelial cells [34,35,36]. LPS also promotes oxidized-LDL/IgM complex binding and foam cell formation by upregulating the expression of Fcα/μ receptors on the surface of macrophages [37]. Additionally, subclinical doses of LPS can reduce interleukin-1 receptor-associated kinase M (IRAK-M) in mice and induce increased expression of miR-24 in monocytes, thereby disrupting the low-grade inflammatory phenotype of monocytes and promoting AS development [33,38]. It has been demonstrated that LPS accelerates atherosclerotic plaque formation through TLR4 by promoting lipid accumulation and the production of chemokines such as MCP-1 in human extravascular fibroblasts via the TLR4-dependent pathway [39]. In contrast, inhibition of Sema3A, a specific membrane-associated secretory protein, weakens LPS-stimulated inflammatory factor release by suppressing the NF-κB and MAPKs signaling pathways, alleviating LPS-induced oxidative stress, and reducing IL-1β, IL-6, NLRP3, SOD-1 antioxidant protein, and reactive oxygen species. This suggests that inhibition of Sema3A could be a potential therapeutic strategy for treating AS [40]. In summary, the activation of TLR4 by LPS and its downstream inflammatory responses play a critical role in the development of atherosclerosis, and targeting this pathway may be a promising therapeutic strategy for treating this disease (Figure 2).

#### 1.3.2. Trimethylamine Oxide (TMAO) with AS

##### Production and Metabolism of TMAO

Trimethylamine oxide (TMAO) is a metabolite produced by intestinal bacteria that has been linked to the progression of atherosclerotic lesions and atherothrombosis [41,42,43]. TMAO is derived from trimethylamine (TMA), which is found in foods containing choline, phosphatidylcholine, and L-carnitine, such as red meat, dairy products, eggs, some fruits, vegetables, and cereals [44]. Gut microorganisms promote the release of TMA precursors from these foods and facilitate the formation of TMA through various microbial enzymes such as Choline TMA lyase, Carnitine monooxygenase, TMAO reductase, Betaine reductase enzyme, which are then converted to TMAO by flavin-containing monooxygenase-3 (FMO3) [26,45,46]. TMA and TMAO are primarily excreted in the urine via renal filtration [47]. Many types of bacteria can produce TMA, including common human commensal bacteria, such as *Aspergillus*, *Clostridium*, *Shigella,* and *Aeromonas*, as well as non-commensal groups, such as *Burkholderia, Shigella, Vibrio, Campylobacter, Aeromonas* and *Salmonella* [48,49]. The concentration of TMAO in blood is influenced by the balance between dietary precursors, TMA-producing bacteria, and TMA-metabolizing bacteria, as well as the activity and number of FMO3 [50]. As more is learned about the various functions and mechanisms of TMAO, its connection to atherosclerosis and thrombosis becomes increasingly clear. Consequently, TMAO has emerged as a promising target for treating AS (Figure 3).

##### TMAO and Lipid Metabolism

Hypertension is a major risk factor for the development of AS, which is influenced by several factors including inflammation, the sympathetic nervous system, water and sodium retention, and glycolipid metabolism [51]. The gut microbial metabolite TMAO has been shown to contribute to the progression of hypertension and AS by inhibiting cholesterol metabolism [52]. TMA, FMO3, and TMAO play important roles in regulating lipid metabolism in the body [53]. Studies have shown that mice with knockout FMO3 fed with a high-cholesterol diet exhibited a decline in intestinal lipid absorption and hepatic cholesterol production, and an increase in cholesterol reverse transportation, thus facilitating the restoration of cholesterol homeostasis [54]. Knocking out coiled-coil 1 in mice resulted in elevated plasma TMAO levels, leading to lipid deposition in atherosclerotic plaques, and increased plasma lipid levels, as well as impaired hepatic cholesterol transportation [55]. Feeding *male apoE^−/−^* mice with 0.3% trimethylamine oxide for eight weeks resulted in a significant increase in the total aortic plaque area and in the elevation of serum triglyceride, total cholesterol, and LDL-c levels in mice [56]. TMAO also affects glucose metabolism in vivo, which directly affects the progression of atherosclerosis [57]. A selective hepatic insulin resistance rat model showed a significant upregulation of plasma TMAO levels and hepatic FMO3 levels. *FMO3 knockout* mice had a lower rate of obesity and atherosclerosis than wild-type mice after high-fat diet induction [49,57,58]. These findings suggest that TMAO may be an emerging target for the treatment of atherosclerosis and related metabolic disorders (Figure 3).

##### TMAO and the Inflammatory Response of the Arterial Wall

Cholesterol-rich macrophages that infiltrate the arterial wall contribute to the development of AS through oxidative stress and inflammation [59]. Through investigating miRNAs and their target sites related to lipid metabolism and cardiovascular expression in TMAO metabolism-related macrophage and stem cell models, it was discovered that TMAO promotes inflammation and AS by upregulating miR-30c-5p and miR-21-5p and downregulating the target gene of *miRNAs*, *Period2 (PER2)*, which controls lipid metabolism and the inflammatory response [60]. Additionally, several studies have shown that TMAO activates scavenger receptors and CD36 on the surface of macrophages, promoting recognition and phagocytosis of oxidized LDL via the CD36/MAPK/JNK pathway, leading to foam cells formation and accelerating the progression of AS [61,62]. Elevated plasma TMAO levels increase the expression of pro-inflammatory cytokines, such as IL-1β, IL-18, and TNF-α, while decreasing the expression of anti-inflammatory cytokines, such as IL-10 [63,64,65]. This process is regulated by TMAO activation of human umbilical vein endothelial cells (HUVECs) oxidation and the ROS/TXNIP/NLRP3 inflammatory vesicle pathway [66]. It was found that TMAO-treated hepatocytes (TMAO-Exos) significantly reduced cell migration and angiogenesis in vitro and impaired endothelial cell function by downregulating CXR4 expression, which affects the recovery of post-ischemic perfusion and angiogenesis [67] (Figure 3).

##### TMAO and Platelet Reactivity

Platelet activation and aggregation are important factors of atherosclerotic thrombosis as enhanced platelet reactivity is closely linked to thrombogenesis, which in turn can result in hypoxic damage and a poor prognosis for the affected tissue [68]. However, there is a debate over whether the intestinal microbial metabolite TMAO actually enhances platelet reactivity and promotes thrombosis. While it has been shown that prothrombotic and pro-atherothrombotic effects are important mechanisms of TMAO’s detrimental impact, there is evidence to suggest that the correlation between TMAO and platelet reactivity is not particularly strong [69,70]. Furthermore, studies have demonstrated that choline and TMAO activate the NOX/ROS/Nrf2/CES1 pathway, which decreases the formation of clopidogrel active metabolite and impairs platelet response to clopidogrel [71]. Despite this, elevated TMAO levels have been significantly associated with coronary CVD in patients who received antiplatelet therapy for CVD [72].

#### 1.3.3. Effect of Gut Microbial Metabolites SCFAs on AS

##### Production and Metabolism of SCFAs

Short-chain fatty acids (SCFAs) are produced in humans by the microbial glycolysis of carbohydrates, such as dietary fiber and resistant starch that are not digested and absorbed, which concludes formate, acetate, propionate, and butyrate [73]. In addition, small amounts of branched-chain fatty acids are produced from the fermentative of protein-derived branched-chain amino acids [74]. The acetate production pathway is widely distributed in multiple floras, while the propionate and butyrate production pathways are highly conserved and substrate-specific [75]. Numerous studies have demonstrated the anti-inflammatory, antihypertensive, and metabolic regulatory effects of SCFAs, which can have a protective effect on the development of AS [76,77,78,79].

##### SCFAs and the Inflammatory Response of the Vessel Wall

SCFAs have been shown to inhibit the development of AS by reducing the migration and activation of macrophages. This effect is achieved by inhibiting the production of cytokines and chemokines [76]. For instance, when butyrate was added to the diet of *apoE^−^/^−^* gene, NF-kappa B translocation was inhibited and the production of *chemokine ligand-2 (CCL2), vascular cell adhesion molecule-1 (VCAM-1)* and *MMP2* gene on chromosome 16 in lesion area was reduced. As a result, the migration of macrophages was decreased, deposition of collagen was increased and plaque stability was improved at the lesion site. In addition, the uptake of oxLDL and production of CD36, TNF, IL-1β, and IL-6 were also reduced, leading to a 50% decrease in the development of AS in the aorta [77,78]. Furthermore, propionic acid and butyric acid have been shown to antagonize the production of LPS-induced cytokines IL-6 and IL-12p40 in humans [79]. This suggests that SCFAs have the potential as therapeutic agents for the treatment of AS and other inflammatory diseases (Figure 4).

##### SCFAs and Blood Pressure Regulation

It is well established that hypertension is one of the major risk factors for the development of AS, and SCFAs have been found to play a protective role against its progression by the regulation of hypertension [77]. Studies have shown that higher concentrations of SCFAs in human feces are associated with higher blood pressure and that hypertensive individuals have a lower number of SCFA-producing microorganisms in their gut microbiota compared to healthy controls [80]. SCFAs regulate blood pressure by targeting different receptors, including G protein-coupled receptors, olfactory receptor 78, and short-chain fatty acid receptors such as FFAR-2 and FFAR-3. For instance, SCFAs acting on FFAR-2 in the renal artery cause arterial diastole and lower blood pressure, whereas SCFAs operating on Olfr78 promote the release of renin and increase blood pressure [81,82,83,84]. Additionally, SCFAs modulate cardiac contractility and sympathetic tone, with mice injected with acetic acid showing a simultaneous decrease in arterial pressure and heart rate, as well as a load-independent decrease in myocardial contractility. These effects were antagonized by beta-1-adrenergic receptor antagonists such as atenolol and tyramine [85] (Figure 4).

##### SCFAs and Adipose Tissue

SCFAs have also been found to directly or indirectly participate in adipogenesis, catabolism, and inflammatory response, leading to inhibition of the progression of AS [86]. For example, propionic acid promotes leptin secretion from the greater omental adipose tissue and subcutaneous adipose tissue by activating GPR41 and GPR43, resulting in significantly increased leptin levels. Moreover, SCFAs have been found to reduce human blood triglyceride concentrations by regulating adipogenesis and impairing lipolysis [87,88,89]. Finally, SCFAs have been shown to inhibit the inflammatory response in adipocytes and adipose tissue, which further prevents the development of cardiovascular diseases, including AS [90] (Figure 4).

#### 1.3.4. Association of Intestinal Microbial Interference with Bile Acid Metabolism and AS

Hydrophobic cholesterol is hydroxylated by cholesterol 7 alpha-hydroxylase (CYP7A1), which is catalyzed in hepatocytes and transforms into hydrophilic primary bile acids [91]. After primary bile acids are excreted through the bile duct, most of which are reabsorbed at the end of the ileum via bile acid transport proteins, some primary bile acids enter the colon and are changed into hydrophobic secondary bile acids through the removal of nuclear hydroxyl groups, oxidation, or epimerization [92], which are reabsorbed by the colon into the circulation and complete the enterohepatic circulation of bile acids. Only about 5% of secondary bile acids are excreted in feces [93]. Bile acids influence the composition of the intestinal flora, and the intestinal flora determines the distribution of intestinal bile acids [84] (Figure 5).

Secondary bile acids have been shown to act as potent signaling molecules that exert different effects by activating the corresponding receptors such as farnesoid X receptor (FXR) and Takeda G protein-coupled receptor 5 (TGR5; GPBAR1). Other receptors that are activated directly or indirectly include sphingosine-1-phosphate receptor 2 (S1PR2), pregnane X receptor (PXR), constitutive androstane receptor (CAR), vitamin D receptor (VDR), and retinoic acid-related receptor γt (R O Rγt) [94]. The activation of these receptors has been linked directly with the development of inflammatory bowel disease, which has been associated with an increased risk of atherosclerosis, cerebrovascular accidents, and atrial fibrillation [95]. Studies have also shown that TGR5 agonist INT-777 has immunosuppressive effects that include reducing macrophage production of pro-inflammatory cytokines, and delaying atherosclerotic plaque formation in *LD^−/−^* mice [96]. Researches have shown that polyphenol-rich substances reduce plasma TMAO by regulating intestinal flora, thereby affecting the development of AS [97]. The earlier confirmed extract was resveratrol (RSV), which has been shown to increase levels of *Lactobacillus* spp. and *Bifidobacterium* spp. in the intestine through intestinal flora remodeling. It also enhanced bile salt hydrolase activity, decreased ileal bile acid content, inhibited enterohepatic foresaid X receptor-fibroblast growth factor 15 (FGF15) axis, increased cholesterol 7a-hydroxylase (CYP7A1) expression, and promoted hepatic bile acid synthesis. All of these diminished the atherosclerosis-promoting effects of TMAO [98]. A recent study has reported that geraniin may also inhibit the progression of atherosclerosis by reducing plasma TMAO concentrations in mice [61] (Figure 5 and Figure 6).

#### 1.3.5. Altered Intestinal Microecology Induced by Increased Intestinal Permeability and AS

The intestinal barrier system, which includes physical, biochemical, and immune components, interacts with intestinal microorganisms and increases intestinal permeability leading to the translocation of intestinal bacterial DNA, as well as promotes the absorption of intestinal flora metabolites and endotoxins into the circulation, thereby accelerating the progression of AS.

The intestinal epithelial cells (IECs) are interconnected, and the symbiotic bacteria in the intestinal lumen secrete antimicrobial substances to inhibit the growth of pathogenic bacteria [99]. The intestinal microenvironment is composed of the glycocalyx, mucus, and water layers, all of which together form the intestinal physical barrier [99,100]. There are three types of intercellular junctions between intestinal epithelial cells: zonula occludens (ZO), zonula adhesion, and bridging granules, which together constitute the apical linkage complex [101]. The tight junction protein (TJ) is involved in regulating epithelial barrier function and intercellular transport, and its regulatory effect leads to the formation of two different paracellular epithelial permeability pathways in intestinal epithelial cells, which are known as the “leakage” pathway and the “pore” pathway, and they are critical factors in controlling intestinal permeability [100,102]. The mucus layer in the intestinal lumen consists of a firmly adherent inner layer and a loosely adherent outer layer; the inner layer contains fewer bacteria and more antimicrobial substances, such as defensins and lysozyme; and the outer layer includes more bacteria and bacterial products, both of which together serve to prevent antigens, toxins, microbial metabolites, and bacteria directly contacting the intestinal epithelial cells. Two layers are essential parts of the physical defense of the intestinal mucosa [103]. The primary participating cells include cup cells and Pan cells [103]. Other cells, which include endocrine cells that secrete GLP-2 and participate in the regeneration and repair of epithelial cells and induce tight protein synthesis, cluster cells that produce IL-25 and IL-13, and M cells that participate in the mucosal immune response, all play essential roles as components of the intestinal mucosal barrier [100]. Intestinal barrier disruption and increased mucosal permeability promote the transport of LPS through the intestinal barrier, which facilitates the involvement of LPS participation in the development of inflammatory responses and AS in vivo. Moreover, LPS affects the tight junction permeability between intestinal epithelial cells through a TLR4-dependent mechanism, which increases the absorption of LPS, bacteria, and metabolites of the intestinal flora (e.g., TMAO, SCFAs, etc.) [34]. It has been shown that one of the ways that daily dietary structure affects the progression of AS is by altering intestinal mucosal permeability, for example, excessive sugar intake and hyperglycemia disruption in the intestinal barrier, which increases intestinal permeability and leads to deregulation of the intestinal microenvironment, thereby leading to an increased incidence of cardiovascular disease [104]. However, the mechanisms of their interaction with intestinal microecology need to be further investigated.

## 2. Intestinal Microecological Disorders Regulation and Atherosclerosis Prevention

Assessment of the composition of gut microbial is an essential foundation for understanding the relationship between gut microecology and AS and is also a prerequisite for developing relevant control strategies. Since most intestinal flora belong to prokaryotes, their 16S rRNA genes, which encode rRNA that is highly conserved and specific, are widely used for researching the composition and distribution of microbial communities through 16S rRNA gene sequencing [105]. In addition, combining 16S rRNA gene sequencing with enzyme-linked immunosorbent assay and lipid metabolism assessment further evaluates the association between the inflammatory responses caused by intestinal microecological disorders, abnormal lipid metabolism, and AS [106]. Based on the results of these tests, the balance of the intestinal can be adjusted to inhibit the development of AS [105,106]. Currently, several strategies have been applied, including intestinal flora transplantation, probiotics, prebiotics, symbiotics, and short-term antibiotic applications, all of which can also help to adjust the microecology of intestinal disorders.

## 3. Discussion

Intestinal flora transplantation is performed by extracting and pretreating microbial communities from healthy feces stool and then transplanting them into the recipient’s colon in order to create a new dynamic balance of intestinal microecology in the recipient’s intestinal microecology [107]. It has been demonstrated that fecal microbiota transplantation (FMT) is effective in the treatment of intestinal bacterial infections, depression, and metabolic diseases such as type 2 diabetes and obesity [108,109,110]. In the practical application of this method, there are still some factors that lead to unstable efficacy, such as the safety and stability of the microbial composition of the donor’s gut, the degree of preparation in the recipient’s gut, the method and standardization of the transplantation operation, etc [107,111]. Therefore, the use of animal models for relevant efficacy assessment is essential for the application of this method. Moreover, prebiotics, probiotics, and postbiotics, such as short-chain fatty acids, can modulate gut microbiota composition and reduce LPS levels, thereby exerting a protective effect against AS [112]. Probiotics are originally referred to as the beneficial flora in yogurt; they affect health by stimulating intestinal flora, interfering with the host immune response, reducing cholesterol absorption, etc [113]. Prebiotics are defined as a particular fermentation component that promotes the growth of probiotics in the human intestinal tract and the function of the body’s immune system; thus, prebiotics are often used as a health food, and lactulose is widely used as a common clinical preparation [114]. Synbiotics are a combination of probiotics and prebiotics, which have characteristics of both and have a more significant and lasting effect on the adjustment of intestinal microecology [115].

The use of inhibitors that reduce TMAO production has been shown to attenuate the promotion of lipid metabolism and inflammation formation by TMAO in the intestinal flora [116,117]. RSV and broad-spectrum antibiotics such as ciprofloxacin can remodel the flora, inhibiting TMAO production by reducing TMA [98]. RSV also attenuates TMAO-induced AS by reducing bile acid (BA) de novo synthesis in the liver [49,98]. The choline analog 3, 3-dimethyl-1-butanol has also been used to reduce TMAO production in the body, which not only weakens the promotion of TMAO in the development of AS but also plays an essential role in the treatment of hypertension and heart failure [118,119]. In addition, adjusting the daily diet structure by reducing the intake of choline-rich foods, such as red meat and seafood, and increasing the intake of plant-based foods, is important in reducing the production and absorption of TMAO in the intestine [120].

Gut microecological disorders affect the development of AS through multiple pathways, and elucidating the corresponding pathways helps to develop a wide range of potential therapeutic targets, which provide new therapeutic modalities and support for clinical mitigation of the development of AS.

## 4. Conclusions and Future Directions

Changes in intestinal microecology mainly refer to alterations in the composition of intestinal microorganisms and the production or absorption capacity of related metabolites, which affect the development of AS by exerting corresponding effects on human immunity, metabolism, and food breakdown. This review details the protective or promotive effects of microbial constituents LPS, microbial metabolites TMAO and SCFAs, microbial interference with bile acid metabolism, and organismal intestinal barrier disruption on AS. This information provides a new idea and direction for the development of clinical prevention and treatment strategies for AS. Numerous studies have demonstrated the relevance of intestinal microecology to the development of AS through multiple pathways. However, research is mostly limited to animal studies, and their feasibility and generalizability in the population still need further exploration. In addition, the current therapeutic measures, such as intestinal flora transplantation, TMAO inhibitors, probiotics, and prebiotics, still have disadvantages including high cost and unstable efficacy, which need further improvement.

## Figures and Tables

**Figure 1 microorganisms-11-01223-f001:**
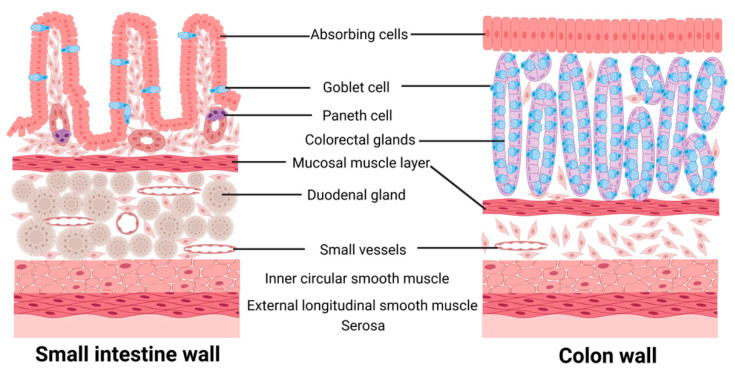
Structure of the intestine.

**Figure 2 microorganisms-11-01223-f002:**
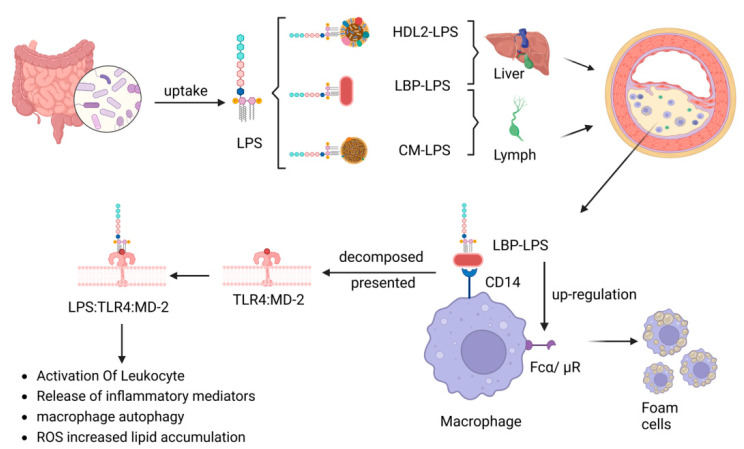
LPS promotes atherosclerotic pathways.

**Figure 3 microorganisms-11-01223-f003:**
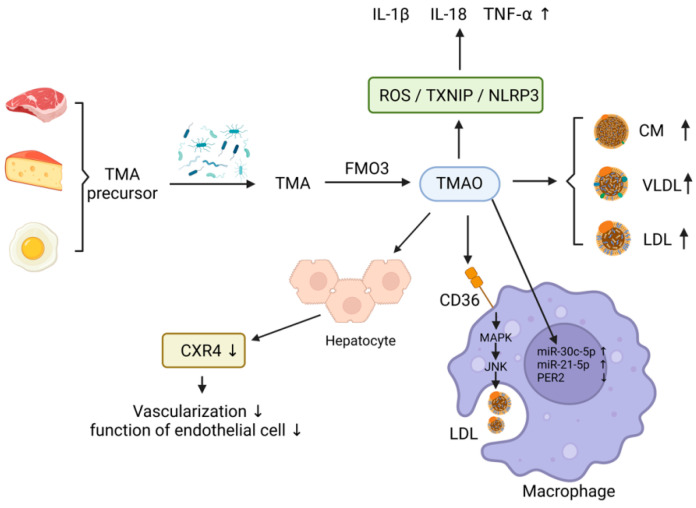
TMAO promotes atherosclerotic pathways. ↑: increase; ↓: decrease.

**Figure 4 microorganisms-11-01223-f004:**
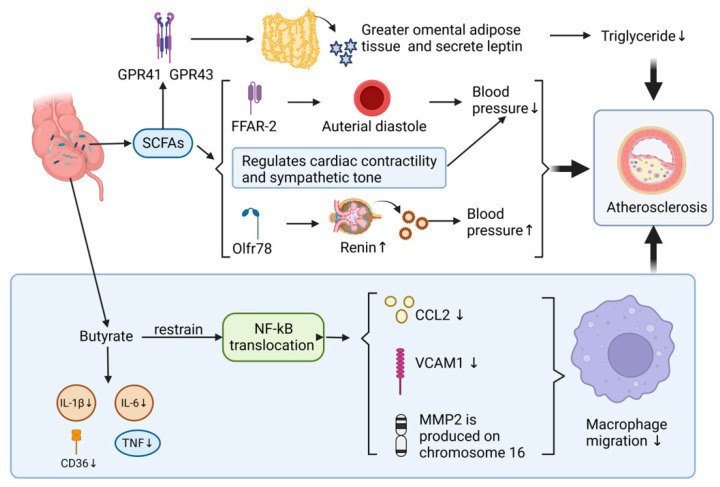
Effect of gut microbial metabolites SCFAs on AS. ↑: increase; ↓: decrease.

**Figure 5 microorganisms-11-01223-f005:**
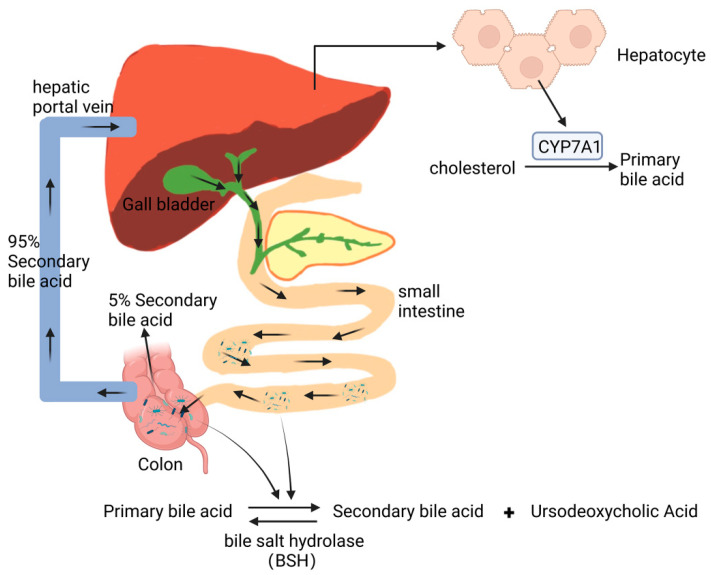
Effect of intestinal microecology on bile acid metabolism.

**Figure 6 microorganisms-11-01223-f006:**
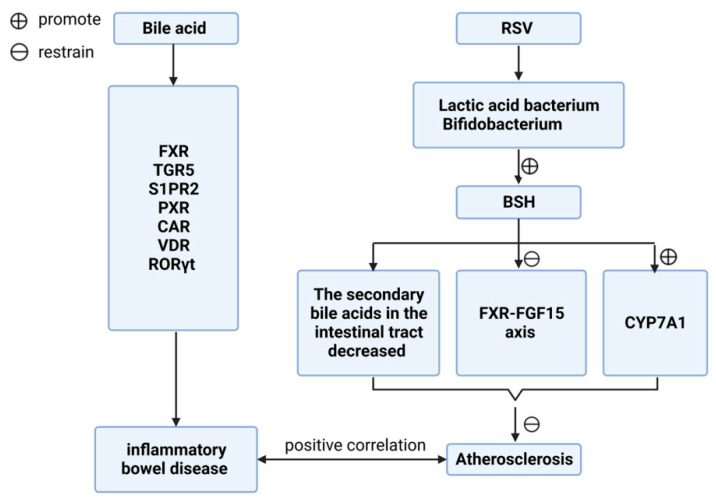
Effect of bile acid metabolism on atherosclerosis.

**Table 1 microorganisms-11-01223-t001:** Common omics techniques and their application.

Common Omics	Application
16S rRNA sequencing analysis	1. Diversity analyses of microbial communities. 2. Abundance and density of species in microbial communities. 3. Interrelationships between microorganisms and hosts.
Macrogenomics	1. Genome assembly analysis of microbial communities. 2. Identification of microorganisms down to species level or even strain level.
Proteomics	1. Protein structure and activity of microbial communities. 2. Complementary or corrected genomic data. 3. Less influenced by the extraction process and more stable than RNA sequencing.
Metabonomics	1. Dynamic changes of metabolites in gut microbiota. 2. Screening for differential metabolites of intestinal flora.
Lipidomics	1. Effect of altered microbiota on lipid metabolism. 2. Interrelationships between microorganisms and hosts.
Transcriptomics	1. Expression levels of mRNAs in microbial communities 2. Metabolic potential of microbial communities 3. Metabolically active members of microbial communities

## Data Availability

All supporting data are available within the article.

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
