# Peer review of "Association between Intestinal Microecological Changes and Atherothrombosis"

_microorganisms, 2023, doi:10.3390/microorganisms11051223_

Round 1

Reviewer 1 Report

The manuscript by Zhuo et al. reviewed the influence of the gut microbiota and related products, such as trimethylamine n-oxide and short-chain fatty acids, on the etiology of atherosclerosis. The topic is a worthy subject for review and the various sections are well articulated and informative. The references are sufficient and up-to-date, and the language is acceptable, although the introduction sometimes lapses into a somewhat pedantic style. Furthermore, in my opinion, some references are cited inappropriately. The figures are of acceptable quality.

A few points must be considered.

Paper title. Since in the body of the article also thrombotic processes are mentioned, in addition to atherosclerosis (e.g. page 6, lines 219-231), I suggest to evaluate the possibility of using the term “atherothrombosis”.

Page 3, lines 94-98. I think that in support of the authors’ statement the reference no. 21 is not appropriate, see the article PMID: 29018189 instead.

Page 3, line 101. Claiming that TMAO has become an “essential mediator” of atherosclerosis seems to me rather exaggerated, given that atherosclerosis can occur without TMAO elevation. Perhaps a softer statement would be preferable.

Page 4, lines 136-139. In the two articles cited, I found no trace of increased production of miR-24. Perhaps PMID: 27824038 is more appropriate.

Page 5, lines 178-181. The cited paper refers to pulmonary hypertension, while for systemic arterial hypertension there are more appropriate paper, for instance PMID: 36824355.

Page 7, line 244. I did not understand “metabolic degenerative effects of SCFAs”

Page 9, line 311. The semisynthetic bile acid INT-777 was not used in the experiments described in reference no. 95. The authors may refer simply to “a TGR5 agonist”

Minor points

Abstract. The first three lines should be rephrased, and it would be better to process the entire manuscript for language editing.

Introduction. The first thirty lines provide trivial information about anatomy and could usefully be shortened, as well as the following sentences, relating to the basic atherosclerotic processes.

Page 1, line 35. “Paneth cells” is preferable in an academic publication.

Page 2, line 45: “vary across populations”.

Page 3, line 100. If possible, use less colloquial expressions such as “has gained increased attention”.

Page 3, line 107. The acronym SCAA has never been defined, perhaps the authors meant SCFAs.

Page 3, line 119. Use lipid-A instead of lipoid-A.

Page 4, line 133. Perhaps: “reactive oxygen species (ROS) and reactive nitrogen species (RNS) formation …

Page 8, line 282 and figure 3. Better: “omental adipose tissue”

Minor editing of English language required

Reviewer 2 Report

Comments: This article has a few conceptual issues that should be studied.

1-       The first part of "introduction" should have been clarified more by plotting a diagram/ scheme illustrating the structure of the intestine. This could help the reader to understand well 

2-       The authors did not provide the factors that can change the normal microbial flora to another one and how these factors led to control of the ratio between different microorganisms populations.

3-       Write the abbreviation of LBP, FMT, etc on the first time of their appearances

4-       The mechanism by which gut microbiota can reduce the concentration of SCFAs in blood plasma is still not clear.

5-       Resveratrol is written in a recent review as a therapeutic agent that can improve microbial flora. authors should explain if this advantage is just related to Resveratrol  or any other polyphenolic compounds

The English language is fine 

Reviewer 3 Report

The manuscript presents a well-written and relevant review of intestinal microecology and atherosclerosis. I have some suggestions I would like to address the authors.

1. Abstract: atherosclerosis definition could be more detailed.

2. Lines 49-50: authors mention several omics technologies. A table explaining each of these technologies would be interesting for readers that are not so familiar with the molecular field.

3. I suggest the authors add an abbreviature list with all the proteins/genes mentioned in the manuscript.

4. I congratulate the authors on their figures, but I believe a table with the microorganisms cited in each section would benefit the manuscript.

5. I suggest the authors add a paragraph or a small section regarding external factors that influence intestinal microecology and their role in atherosclerosis, such as age, excessive use of antibiotics, diet, etc.

6. Although the manuscript is comprehensible and well-structured, an English review would be helpful, especially regarding the verbs used.

7. Please italicize the gender/species names and the genes, so they are not confounded with their encoded proteins.

I suggest moderate English revision regarding the verbs and translation issues. Some examples throughout the text:

Line 29: "The human intestine is the digestive tract", I suggest to alter to "is part of the digestive tract".

Lines 47-48: "the growth of relevant sensitive flora", what do the authors mean by "relevant sensitive"?

Line 138: I suggest altering the word "production" to "expression" in regard to miR-24.

Line 204-205: Please check the "period2" word, it might be a misspelling.

Line 214-215: Are the treated hepatocytes that actively reduce the cell migration or do these cells have their migration potential reduced? Please check the verbs.

Round 2

Reviewer 1 Report

Overall, the authors have corrected the manuscript according to my indications and eliminated language inaccuracies as far as possible. Only “Resently” -> recently, would remain to be amended in line 22 of the revised version. In addition, lines 38-44 are a repetition of the following ones and should be fixed. For the rest, I have no other comments to make.

There are only a few inaccuracies that need to be corrected

Reviewer 2 Report

The manuscript has been revised point by point according to reviewer's comments  and It is more acceptable NOW

English  Typos  revised thoroughly